# Copper-Treated Environmentally Friendly Antipathogenic Cotton Fabric with Modified Reactive Blue 4 Dye to Improve Its Antibacterial and Aesthetic Properties

**Muhammad Shahid** [1], **Azam Ali** [1,*] , **Nageena Zahid** [2], **Muhammad Shahzad Anjam** [2] , **Jiri Militky** [1] , **Jakub Wiener** [1] , **Sundaramoorthy Palanisamy** [1] and **Blanka Tomkova** [1]

1   Department of Material Science, Technical University of Liberec, 461 17 Liberec, Czech Republic
2   Institute of Molecular Biology and Biotechnology, Bahauddin Zakariya University,
    Multan 60800, Punjab, Pakistan
*   Correspondence: azam.ali@tul.cz; Tel.: +420-774357957

**Abstract:** The objectives of the present study were to develop an environmentally friendly, low-price, easy, and fast method for developing antipathogenic (antibacterial, antifungal, and antiviral) cuprous-oxide-coated multifunctional fabrics. The fabrics were first sensitized with citric acid, and then $Cu_2O$ particles were formed using the Fehling solution method. The cuprous oxide particles were then applied to the cotton fabrics. To create the $Cu_2O$ particles, three different kinds of reducing agents with varying concentrations were used. SEM, dynamic light scattering, FTIR, EDS, and XRD were used to examine the surface morphologies and metal presences. In the second step, a reactive antibacterial dye was made (by reacting Reactive Blue 4 with triclosan). The molecular structure of the modified dye was confirmed with FTIR. The resultant antibacterial dye was applied on the copper-treated cotton fabrics in accordance with the exhaust dyeing protocol. The dyed fabrics were characterized through the colorimetric data (L*, a*, b*, C, H, and K/S), levelness of dye, fastness properties as well as exhaustion and fixation rates. Cuprous-oxide-coated fabrics were tested for antipathogenic activity using quantitative and qualitative measurement results. The fabrics treated with cuprous oxide particles reduced with sodium hydrosulfite at 1 g/L seemed to have the highest antipathogenic effect. Moreover, the versatility of the hygienically developed bioactive fabrics in terms of their comfort properties such as air permeability and stiffness were investigated. Finally, the coating's durability was confirmed by evaluating its antibacterial properties and performing an SEM analysis after laundry.

**Keywords:** antimicrobial; hospital-acquired infections; medical textiles; cuprous oxide particles; color analysis

## 1. Introduction

Since the COVID-19 pandemic, the prominence of antimicrobial textiles, primarily those used in hospitals, has escalated [1]. Due to being in constant contact with the human body and their high moisture-retentive properties and large surface areas, textile materials provide the appropriate environment for microbial growth and rapid multiplication [2]. The textile materials used in hospitals for clinical application must be non-toxic, non-allergenic, antimicrobial, and anti-inflammatory to comply with high standards and stringent requirements [3]. In the healthcare sector, textiles (such as drapes, upholstered furniture, clothing, etc.) are crucial in the transmission and acquisition of contagious diseases. Due to the growing biological threats to global security, it is imperative to develop antimicrobial textiles for hygienic practices and preventative measures to avoid cross-infections in hospitals.

Conventionally, different natural fibers, such as bacterial cellulose [4], cellulose nanocrystals, starch, lignin, and pectin [5], have been used as bio-based materials to functionalize the textiles. However, their applications in textiles have the limitation of poor washing durability.

Recently, metallic nanoparticles are increasingly being used to functionalize fabrics in an effort to prevent the transmission of disease and bacterial growth [6,7]. The researchers have been utilizing various kinds of nanoparticles for the antimicrobial finishing of hospital textiles. Among them, the nanoparticles of $TiO_2$, Cu, ZnO, Ag, MgO, $Cu_2O$, CuO, etc., are the most prevalent and of specific interest [8–11]. The synthesis and applications of copper nanoparticles for the development of antimicrobial textiles have attracted a lot of attention in the past few years. Various techniques, which include photochemical methods, laser ablation, chemical reduction, gamma irradiation, biological synthesis methods, etc., have been used for the synthesis of Cu-NPs [12–14]. Copper, its alloys, as well as copper ions have shown strong antiviral, antifungal, and antibacterial action toward a wide range of microbes. In recent years, Cu and CuO nanoparticles have garnered a lot of interest due to their potential applications as conductive inks, cooling fluids, heat transfer networks, as well as in the development of antimicrobial textiles [15].

Metallic coatings on inflexible substrates have been made possible via the well-established industrial technology known as electroless deposition [16]. Moreover, its potential for antimicrobial textile development has recently been explored, and electroless depositions of copper, silver, and zinc have been performed with intriguing results for multifunctional textiles [8,17–19]. Ali et al. described the electroless deposition of Cu-NPs nanoparticles on cotton fabric in order to produce multifunctional textiles. The cotton fabric that was developed was highly effective against pathogenic microbes [20]. These researchers also reported making antibacterial textiles by depositing Cu-NPs on cotton fabric. The developed textiles inhibited both Gram-negative and Gram-positive bacterial strains effectively [21]. Shahid et al. investigated the deposition of $Cu_2O$ nanoparticles on cotton fabrics using varying concentrations of reducing agents. The Cu-NP-treated cotton fabric demonstrated outstanding antibacterial activity against test microbes [15]. Excellent antibacterial activities have been attained through the deposition of copper nanoparticles by employing these technologies. However, such treatments have led to some undesirable effects such as discoloration or staining on the coated fabrics, which affect the aesthetic properties of the textiles. In some studies, dyeing of the coated textiles was performed to overcome discoloration and staining but their antibacterial effectiveness was compromised [22]. Thus, the development of highly effective antimicrobial textiles with improved aesthetics is still challenging.

Considering the above-mentioned problems, the current study proposes a novel approach for the development of cuprous-oxide-coated antibacterial cotton fabrics with an excellent aesthetic appearance. Initially, citric acid was used to sensitize the fabrics. The Fehling methodology was then used to produce $Cu^2O$ nanoparticles. The synthesized NPs were then applied on the cotton fabrics. In the second step, a reactive dye was selected and functionalized with an antibacterial agent. Subsequently, the cuprous oxide particle-coated fabrics were subjected to exhaust dyeing with the solution of a functional bioactive dye. Fabrics treated with antibacterial dyes have poor washing durability due to which their application with these dyes is not sustainable. Therefore, the synthesis and durable immobilization of nanoparticles, particularly copper nanoparticles, on textiles has gained considerable attention in recent years due to their excellent washing durability. However, treating a textile with copper or silver nanoparticles significantly alters the hue of the fabric, thus affecting its aesthetic properties. Therefore, in the current study, the nanoparticles were applied on cotton fabrics to achieve durable antimicrobial activity, and the treated fabrics were dyed with the antibacterial dye to maintain their aesthetic as well as antibacterial properties.

The objectives of the present research are as follows:

- The synthesis of cuprous oxide nanoparticles by using three different reducing agents.
- Application of the synthesized Cu-NPs on cotton fabrics to impart antibacterial functionality.
- Application of antibacterial dye on copper-coated fabrics to improve their aesthetic appearance.

As such, we tried dyeing the cuprous-oxide-treated fabrics to make them suitable for commercial applications with enhanced antimicrobial abilities. The end applications of the developed textiles could be in the fabrications of antimicrobial bed sheets, surgical drapes, panels, surgical gowns, pants, panel covers, wallpaper coverings, shoe mats, scrub suits, table coverings, chair coverings, socks for doctors and patients, etc.

## 2. Experimental Section

### 2.1. Materials

The substrate for making the antibacterial materials was a plain-woven cotton fabric with an areal density of "f 220 g/m$^2$". The chemicals used for the synthesis and deposition of cuprous oxide (Cu$_2$O) had 99.99% purity. Table 1 lists the components that were used in this study. Reactive Blue 4 (35% dye content) was purchased from Sigma-Aldrich, Lahore, Pakistan. Triclosan (97%) was procured from TCI, Tokyo, Japan.

**Table 1.** List of materials used in the present study.

| Material Description | Source |
|---|---|
| Plain-woven 100% cotton bleached fabric, areal density 220 g/m$^2$ | Licolor, a.s., Liberec, Czech Republic |
| Sodium potassium tartrate | ACS reagent, Prague, Czech Republic |
| Copper sulfate pentahydrate | ACS reagent, Prague, Czech Republic |
| Ascorbic acid | ACS reagent, Prague, Czech Republic |
| Glucose | Aldrich Reagent-plus, Lahore, Pakistan |
| Na$_2$S$_2$O$_4$ (sodium dithionite) | ACS reagent, Prague, Czech Republic |
| Reactive Blue 4 | Sigma-Aldrich, Lahore, Pakistan |
| Triclosan | TCI Japan, Tokyo, Japan |

### 2.2. Preparation of Cuprous Oxide Particles and Deposition on Cotton

Cuprous oxide particles (Cu$_2$O) were synthesized with combinations of two Fehling (A and B) solutions and three separate reducing agents, namely, glucose, ascorbic acid, and sodium hydrosulfite. Separate preparations of Fehling solutions A and B were made. For the preparation of Fehling A, 69.28 g of CuSO$_4$.5H$_2$O was dissolved in 1 L of distilled water and stirred continuously. Then, a few drops of H$_2$SO$_4$ were added (we added 5 drops). An amount of 350 g of sodium potassium tartrate and 140 g of NaOH were mixed in 1 L of distilled water to prepare the Fehling B solution. The Fehling A and B solutions were mixed with a 1:1 ratio. Then, a reducing agent (glucose) was immediately added to the mixed solution (2.5% of the total weight of the mixed solutions—Fehling A + Fehling B).

The solution of Fehling A and B and glucose was applied to cotton fabric using the padding method (the contact time between the fabric and solution was max. 10 s) on a padder, and the wet allowance was approximately 75%. After squeezing the excess solution from the fabric, the fabric was placed in a heat press for 4 min at 50 °C. After removing the fabric from the press, the sample must be washed, preferably in boiling water, to remove the alkali and then dried. The same process was conducted for the other two reducing agents, i.e., sodium hydrosulfite and ascorbic acid.

### 2.3. Functionalization of Reactive Blue 4 Dye with Triclosan

The functionalization of the Reactive Blue 4 (2) dye with triclosan (1) was carried out in a single-step reaction. An equimolar amount (20 mM) of both reactants was taken. In a round bottom flask, 13.6 g of Reactive Blue 4 dye was added and dissolved in 100 mL of distilled water (solution A). Solution A was refluxed and the temperature was maintained at 40–45 °C. In a 100 mL beaker containing 20 mL of methanol, 5.79 g of triclosan (antibacterial agent) was added and stirred (solution B). Solution B was slowly added to solution A. The pH of the solution was kept neutral, using a 3% $w/v$ solution of sodium carbonate. The progress of the reaction was monitored using thin-layer chromatography (TLC) with an ethyl acetate and petroleum ether (70:30) solvent system. The reaction mass was stirred

at 45 °C until the spot of triclosan on the TLC plate disappeared, which confirmed the successful completion of the reaction and product formation. The functionalized dye (3) was filtered and dried in an oven at 40 °C. The proposed condensation reaction showing the formation of the functionalized dye is given in Figure 1.

**Figure 1.** Antibacterial functionalization of Reactive Blue 4 dye with triclosan.

### 2.4. Application of Functionalized Dye on Fabric

The functionalized reactive dye was applied on the copper-coated cotton fabric using the exhaust dyeing method. For this purpose, an H-T dyeing machine was used. A 3% dye shade (o.w.f) was applied on the fabric with a material-to-liquor (M:L) ratio of 1:50. The dyeing of the fabric was started at room temperature, which was gradually increased to 70–80 °C. An electrolyte (sodium sulfate, 40 g/L) and alkali (1 g/L of sodium hydroxide and 15 g/L of sodium carbonate) were added portion-wise in the dyebath at different time intervals for the proper exhaustion and fixation of the dye. The dyeing of the fabric was continued for 60 min. After that, the dyed fabric was removed from the dyebath and rinsed with tap water followed by hot water that was at 90 °C for 10 min to wash away the unfixed dye molecules from the fabric. The possible formation of covalent bonds between the chlorine atoms of the triazine reactive system of the dye and the hydroxyl groups of the cotton fabric is shown in Figure 2.

**Figure 2.** Schematic showing covalent bond formation between functionalized reactive dye and cotton fabric.

Hence, we developed a total of 6 samples for the 3 reducing agents (3 were dyed and 3 were undyed). The design of the experiment is given in Table 2 below. Figure 3

illustrates the three-step method involved in the synthesis of cuprous oxide particles and their subsequent deposition on cotton fabric.

**Table 2.** Design of experiment for the developed samples.

| No. of Samples | Reducing Agent | Applicant of Dye | Sample Code |
|---|---|---|---|
| 1 | Glucose | No | G |
| 2 | Glucose | Yes | DG |
| 3 | Ascorbic acid | No | A |
| 4 | Ascorbic acid | Yes | DA |
| 5 | Sodium hydrosulfite | No | S |
| 6 | Sodium hydrosulfite | Yes | DS |

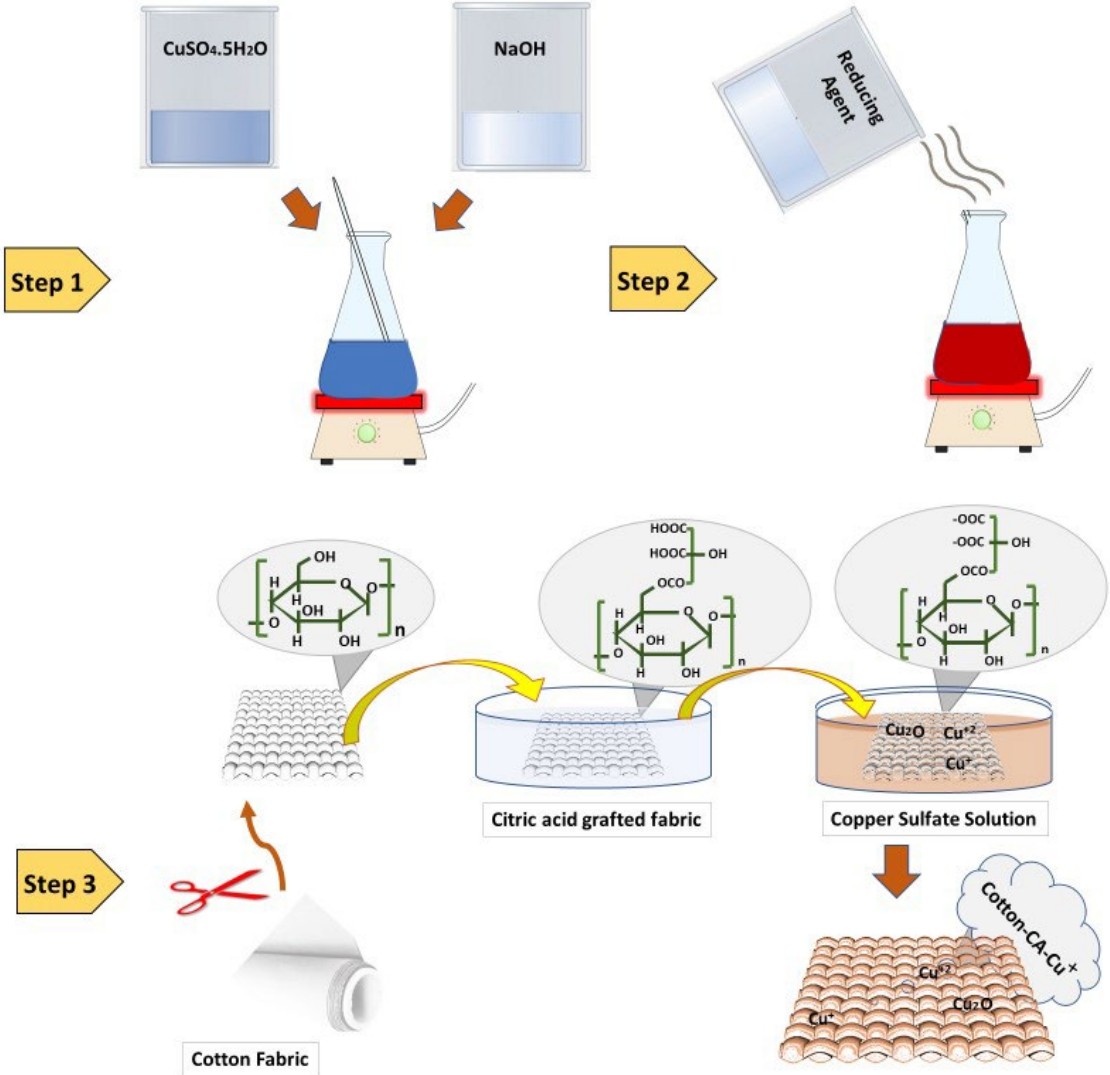

**Figure 3.** Schematic illustration of the three-step method involved in the synthesis of cuprous oxide particles and their subsequent deposition on cotton fabric.

The schematic showing the modification of the dye is shown below in Figure 4.

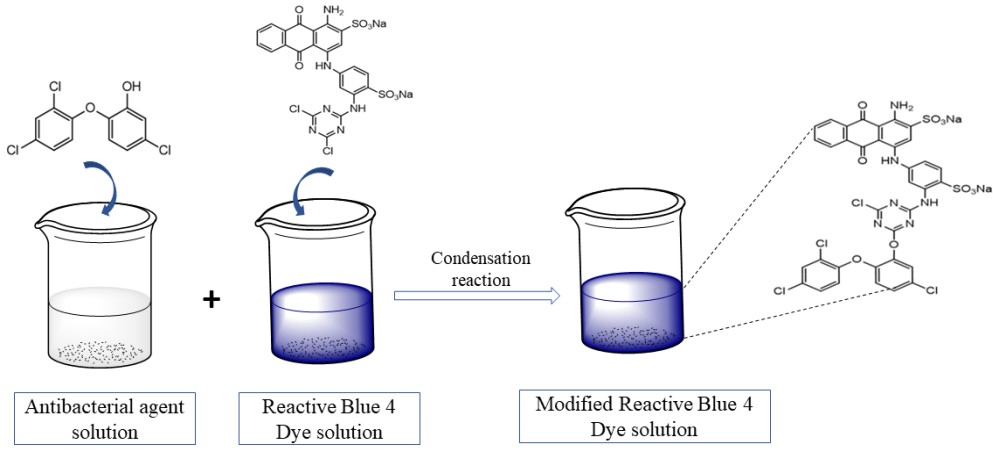

**Figure 4.** Schematic illustration of the functionalization of dye.

The schematic showing the application of the modified dye on the copper-treated fabric is shown in Figure 5.

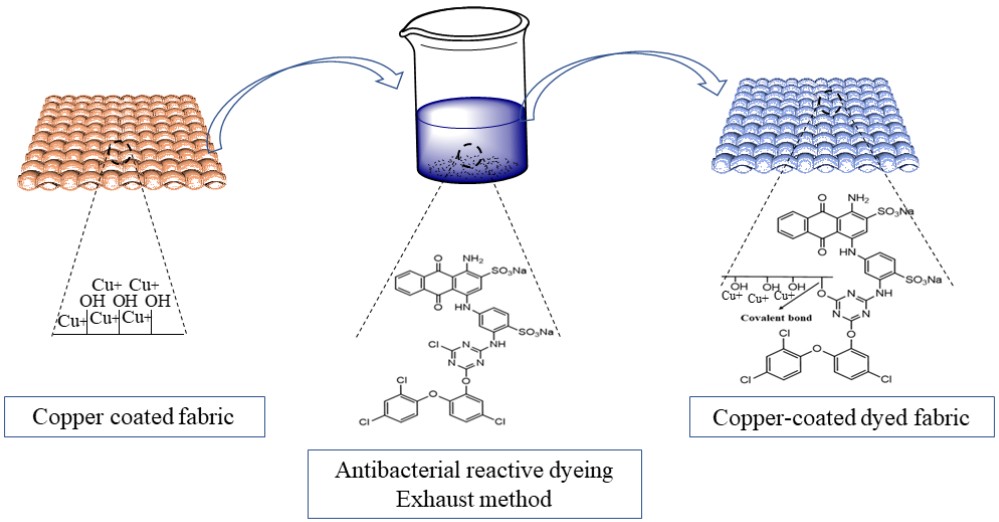

**Figure 5.** Schematic illustration of the subsequent dyeing of $Cu_2O$-coated fabrics.

*2.5. Characterization*

2.5.1. Surface Characterizations

The morphological characteristics of the $Cu_2O$ nanoparticles coated on the surface of the cotton fabric were examined using SEM with a Quanta 50, FEI (Liberec, Czech Republic), X-ray diffraction (XRD) characterization, and the dynamic light scattering (DLS) theory with a Malvern Zetasizer from Panalytical X'pert PRO (Liberec, Czech Republic) tools. The identification of the functional group of the modified reactive dye was performed using the FTIR spectrum with an FTIR Perkin-Elmer (Liberec, Czech Republic) spectrophotometer.

2.5.2. Dye Exhaustion (E%), Fixation (F%), and Total Fixation (T%) Analysis

The exhaustion, fixation, as well as total fixation rates for the modified dye were determined using the method described in [23], and the following equations were employed for the calculations:

$$\%E= [1 - C_2/C_1] \times 100 \tag{1}$$

Here, $C_1$ and $C_2$ represent the concentration of the dye calculated after as well as before dyeing:

$$\%F = [(C_1 - C_2 - C_3)/C_1 - C_2] \times 100 \tag{2}$$

and C$_3$ represents the concentration of the extracted dye.

$$\%T = (\%E \times \%F) \times 100 \tag{3}$$

Meanwhile, F, T, and E indicate the fixation, total fixation, and exhaustion of the dye.

### 2.5.3. Fastness Properties

The light, rubbing, and washing fastness properties of the copper-treated dyed fabric samples were assessed. The ISO 105-X12, ISO 105-C06, and ISO 105-B02 standards were followed for the evaluation of rubbing fastness, washing fastness, and light fastness, respectively.

### 2.5.4. Colorimetric (CIELAB) values analysis

The CIELAB values (a*, b*, h*, L*, C*) and K/S for the copper-treated undyed and copper-treated dyed fabric samples were determined using a reflectance spectrophotometer. Negative and positive values of b* indicate the degree of blueness and yellowness of a dye, respectively, whereas negative values of a* indicate the extent of greenness and positive values the degree of redness. Chroma is represented by C*, brightness by L* (values ranging from 0 to 100, where 0 represents a pure black color and 100 is the value of a pure white color), and h* represents the hue angle (00–3600). The K/S values for dyed fabrics were estimated employing Kubelka–Munk Equation (Equation (4)):

$$K/S = (1 - R)^2 / 2R \tag{4}$$

Here, R indicates the percentage reflectance, K the absorption coefficient, and S the scattering coefficient.

### 2.5.5. Assessment of Dye Levelness

Both a visual and an objective technique were used to assess the dye levelness of the dyed fabric with a modified reactive dye. For the visual examination, the fabric was observed carefully from various angles of the colored fabric, and grading from 1 to 5 was carried out (5 for excellent levelness, whereas 1 for poor levelness). The degree of levelness was assessed using an objective method for more precise findings. To do this, the fabric was detected using a reflectance spectrophotometer at 12 separate points, and K/S values were calculated. The calculated standard deviation for each K/S measurement was utilized to interpret the dye levelness [18]. Higher dye levelness is correlated with lower standard deviation values; values in the range of 0.20 indicate excellent dye levelness, while values of >1.0 indicate poor dye levelness. The standard deviation was calculated using Equation (5) for the measured K/S values:

$$S.D = \sqrt{\frac{\sum(X - \bar{x})^2}{n - 1}} \tag{5}$$

Here, X represents the K/S for each scan, $\bar{x}$ indicates the mean value for all measured K/S readings, and n indicates the total number of scans.

### 2.6. Antimicrobial Properties

The antibacterial potentials of the copper-treated undyed and dyed fabric samples were tested both qualitatively and quantitatively using standard testing protocols.

### 2.6.1. Zone of Inhibition Test
Preparation of Bacterial Strain

The Gram-positive S. aureus (CCM 3953) and Gram-negative E. coli (CCM 3954) strains of bacteria utilized in this investigation were acquired from Masaryk University, Czech Republic. Fresh bacterial cultures were mostly prepared by culturing a single

colony in a nutrient broth overnight at 37 °C. The sample turbidity was normalized to an optical density of 0.1 at 600 (OD600) prior to conducting the antibacterial assays. Each agar plate was freshly prepared before the antimicrobial tests. The cells were evenly distributed throughout the agar plates after being inserted into the culture suspension with a sterile cotton swab. The samples were immediately loaded on prepared plates for further investigation.

Determining Zone of Inhibition

A detailed description of the technique is provided by Padil, Nguyen, 2015 [18]. We measured the antibacterial activity of the copper-coated undyed and copper-coated dyed fabric samples, of which $3 \times 3$ mm squares were loaded onto the infected agar plates. The untreated cotton fabric was tested as a control. The agar plates loaded with fabric samples were incubated at 37 °C for 24 h. The ZOI was calculated, which represents the total diameter (mm) of a coated textile sample along with the halo zone where no bacterial growth was observed. The experiment was performed in triplicate and a mean value was taken.

2.6.2. Reduction Factor (Quantitative Test)

For the quantitative anti-bacterial assessment of the dyed cotton, the ISO 20743:2013 transfer protocol was used. As directed in the method, the agar media plates were prepared and infected with 1 mL of cell suspension. The control (2 samples) and dyed swatches (2 samples) were positioned on the prepared agar surfaces, and specimens were pressed down with a 200 g weight. One test sample from each sample was removed from the agar surface and positioned on a separate Petri plate with the transferred surface facing up. The specimens were then cultured for 24 h at 37 degrees Celsius. The second specimens of the control and dyed samples were immediately transferred to 2 separate reagent plastic containers with 30 mL of saline (0.85% NaCl) to determine the bacterial culture count at 0 hr. After shaking the containers for 10 min, 9 dilutions of this saline were prepared, and all dilutions were plated on agar growth media according to ISO 20743. The same procedure was followed for the samples that were placed in an incubator for 24 h to determine the bacterial colony count. Equation (6) was used to calculate the antibacterial activity (A) of the dyed cotton sample. Each sample was run in triplicate to ensure that the results were correct.

$$A = F - G \tag{6}$$

Here, F= (log $C_t$ − logC$_0$), $C_0$ and Ct is the colonial count of the control at 0 and 24 h, G= (log $T_t$ − log $T_0$), and $T_0$ and Tt is the colonial count for the treated fabrics at 0 and 24 h.

2.6.3. Antifungal Analysis

The antifungal property of the coated and dyed fabric samples was assessed using the AATCC 100-2004 standard testing method. *Aspergillus. Niger*, a fungus species, was used for this test. Equation (1) was used to determine the antifungal effectiveness in terms of the percent change.

$$\text{Percentage reduction R}(\%) = \frac{(A - B)}{A} \times 100 \tag{7}$$

Here, A and B indicate the fungal spore counts for the dyed and control fabrics, respectively.

2.6.4. Antiviral Activity

The determination of the virus titer reduction from the initial viral titer of infectivity (107) titer was performed using Behren and Karber's method. Vero-E6 cultures were maintained in Dulbecco's Modified Eagle Medium, which contained 2% penicillin-streptomycin and 9% fetal-bovine serum (FBS) (PSA). The Vero-E6 cultures were infected with the coronavirus with a ratio 1:3 in polyethylene containers, and virus strains developed after one day.

The virucidal impact of the created viral stocks was investigated under a microscope. An amount of 10% FBS was added to the cell line, which was then frozen at 90 °C. Moderate centrifugation was used to filter the supernatant for 30 min at 5–7 °C and at 3700 rpm. The supernatant was used as the viral stock in the experiment, and all the macro residual was eliminated. The Vero-E6 cell lines were deposited at a concentration of $2 \times 10^5$ in 96-well plates and cultured under normal conditions (24 h at 37oC in 6% $CO_2$) to determine the virus titers. Each sample was diluted ten times from $10^1$ to $10^8$. Every dilution was injected into cell lines, where they were cultured for 3 days at 6% $CO_2$. The procedure established by Behren and Karber was used to measure the coronavirus titers in the cultivated cell lines. Following that, $20 \times 20$ mm fabric sample vials were filled with the treated and control fabric samples. An amount of 100 μL of viral loads was passed through the treated and control fabrics, and any recoverable viral loads in containers were cleaned with the filter. There was a $10^1$ to $10^8$ dilution of the coronavirus. All serial dilutions were implanted into the Vero-E6 cell lines, where they were then cultured for 3 days at 37 °C with 6% $CO_2$. The method of Behren and Karber was employed to determine the coronavirus titers in the cultivated cell lines.

### 2.7. Durability of Bioactive Fabrics

The durability of the developed fabric samples was evaluated to check their stability in service. The fabrics were washed in accordance with ISO 105-C01. All fabric samples were mixed in a conventional detergent solution with a 50:1 liquor ratio. After that, the samples were washed for 35 min at 40 °C at a 600 rpm speed. The fabrics were then dried and conditioned for a total of 24 h under normal atmospheric conditions. Electrical conductivity, antibacterial findings, and SEM observations all supported the durability.

## 3. Results and Discussion

### 3.1. FTIR Analysis

The FTIR Peaks (KBr): 3323 $cm^{-1}$ (amine –$NH_2$, –NH stretch), 3378 $cm^{-1}$ (amine –$NH_2$, N–H stretch), 1046 $cm^{-1}$ (C–O–C ether linkage stretch), and 638 $cm^{-1}$ (C–Cl stretch). The FTIR spectra of the unmodified dye (Reactive Blue 4) and modified dye (modified with an antibacterial agent) are shown in Figure 6. The spectra successfully confirmed that the antibacterial agent (triclosan) was successfully incorporated into the structure of the dye through covalent bonds. It was confirmed by the existence of a sharp peak of strong intensity at 1046 $cm^{-1}$. This sharp peak is a characteristic peak of the ether linkages (C–O–C) [24]. The formation of a covalent bond between the hydroxyl group of triclosan and chlorine of the triazine ring resulted in the development of an ether linkage (C–O–C). Therefore, the absence of a sharp and strong peak at 1046 $cm^{-1}$ in the spectrum of the unmodified dye and the appearance of this peak in the spectrum of the modified dye confirmed that the modification of the dye with an antibacterial agent was successfully achieved [25,26].

The modification of the dye was further supported by the increase in the intensity of the peak that occurred at 638 $cm^{-1}$. This peak was also present in the spectrum of the unmodified dye and is attributed to the C–Cl stretching vibrations [27,28]. However, the intensity of the same peak was significantly increased in the spectrum of modified dye. The increase in peak intensity could be due to the increase in the C–Cl linkages in the structure of the modified dye because three C–Cl linkages are present in the structure of triclosan, which further confirmed that triclosan was embedded in the structure of the dye molecule [29].

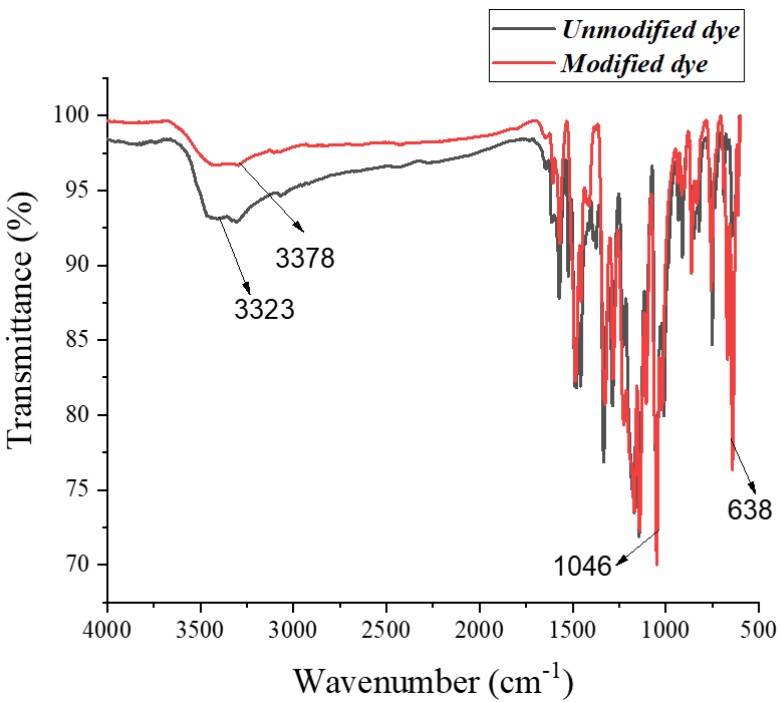

**Figure 6.** IR spectra of modified and unmodified dye.

### 3.2. Colorimetric Data Measurement

The colorimetric data for the copper-treated undyed and dyed fabric samples were evaluated and the obtained results are given in Table 3. There was a significant difference between the colorimetric data for the copper-treated undyed and dyed fabrics. The K/S value for the dyed cotton fabric was higher (12.13) compared with the undyed fabric (7.08), which showed that application of the dye changed the light-colored fabric into a comparatively dark-colored fabric. The dark shade of the dyed fabric samples was further confirmed by the difference in the L* values for both fabrics. The L* value for the dyed fabric sample was lower (39.14) than the L* value for the undyed fabric sample (52.43), which showed that the dyed fabric had a darker shade compared with the undyed fabric. The chroma (C*) value for the dyed sample was also lower (19.14) than that for the undyed fabric (32.87), which suggested the brighter shade of the undyed fabric samples and the darker and duller shade of the dyed fabric sample. The a* and b* values for the undyed fabric samples were positive, which indicated the reddish and yellowish shade of the undyed fabric, whereas both the a* and b* values were negative for the dyed fabric samples, indicating the bluish and greenish shade of the dyed fabric samples.

**Table 3.** Colorimetric data for copper-treated undyed and dyed fabric samples.

| Sr.# | Properties | Copper-Coated Fabrics | Copper-Coated Dyed Fabrics |
|---|---|---|---|
| 1. | Fabric color | | |
| 2. | K/S | 7.08 | 12.13 |
| 3. | L* | 52.43 | 39.14 |
| 4. | a* | 5.15 | −9.34 |
| 5. | b* | 31.56 | −16.13 |
| 6. | C* | 32.87 | 19.14 |
| 7. | H* | 80.15 | 241.13 |

K/S (depth of the color), L* (lightness), a* (red/green value), b* (blue/yellow value), C* (chroma value), and H* (hue value).

### 3.3. Levelness of Copper-Treated Undyed and Dyed Fabric

The color levelness effect was determined for the evaluation of the even appearance of the dyed fabric. For this, a reflectance spectrophotometer was used to scan both the undyed copper-coated and dyed copper-coated cotton fabrics, and K/S values were taken at 12 distinct points. Table 4 displays the acquired K/S values and standard deviation determined from these values. The standard deviation value calculated for the dyed copper-coated fabric was 0.11, which suggested that the dyed fabric had excellent levelness properties and that the dye was evenly present all over the surface of the fabric, whereas, the standard deviation for the undyed copper-coated fabric was 2.16, which revealed the highly uneven appearance of the copper-treated fabric. The results of the dye levelness effect confirmed that the dyed copper-coated fabric had an even and smooth appearance compared with the undyed copper-coated fabric, which was one of the objectives of our study. In the visual evaluation, the dyed copper-coated fabric was assigned grade 5 (excellent levelness), while the undyed copper-coated fabric was given grade 2 (poor levelness), which further supported the above findings, i.e., the application of the dye on the copper-coated fabric produced the even and smooth appearance of the fabric.

**Table 4.** Reflectance measurements for the undyed and dyed copper-coated fabrics.

| Number of Scans | K/S Values Undyed Sample | Standard Deviation (S.D) | K/S Values Dyed Sample | Standard Deviation (S.D) |
|---|---|---|---|---|
| Reading 1 | 12.97 | | 15.24 | |
| Reading 2 | 12.97 | | 18.67 | |
| Reading 3 | 12.95 | | 11.55 | |
| Reading 4 | 12.97 | | 8.73 | |
| Reading 5 | 12.98 | | 16.81 | |
| Reading 6 | 12.97 | 0.19 | 10.78 | 2.16 |
| Reading 7 | 12.96 | | 17.78 | |
| Reading 8 | 12.97 | | 9.77 | |
| Reading 9 | 12.98 | | 14.65 | |
| Reading 10 | 12.96 | | 7.63 | |

### 3.4. Fastness Properties of Copper-Coated Dyed Fabric

The exhaustion, fixation, washing, rubbing, and light fastness properties of all three samples DG, DA, and DS were calculated according to standard testing methods, and the obtained results are provided in Table 5. All dyed samples revealed excellent washing fastness (4–5), excellent light fastness (4–5), and good rubbing fastness (4) properties. The dye exhaustion and fixation rates of all the samples were also significantly high, i.e., 90% exhaustion and >85% fixation. The excellent fastness properties and high exhaustion and fixation rates for the dyed samples could be due to the fiber-reactive nature of the reactive dye. Reactive dyes can form strong covalent linkages with the hydroxyl groups of cotton fabrics, due to which excellent dye fixation and fastness properties can be achieved.

**Table 5.** Exhaustion, fixation, and fastness (washing, rubbing, and light) results of dyed fabric.

| Sr. # | Sample Code | Exhaustion % | Fixation % | Washing Fastness | Rubbing Fastness | Light Fastness |
|---|---|---|---|---|---|---|
| 1. | DG | 91 | 84 | 4–5 | 4 | 4–5 |
| 2. | DA AG | 93 | 85 | 4–5 | 4 | 4–5 |
| 3. | DS SG | 93 | 84 | 4–5 | 4 | 4–5 |

### 3.5. Morphology of Copper-Coated Dyed Cotton Fabrics
SEM microstructure

Scanning electron microscopy for all the cuprous-oxide-coated dyed fabrics was employed to observe the morphological changes with each reducing agent. Figure 7 depicts

nanometer-scale images of the cuprous oxide particle surface morphologies on the surface of the colored fabric. The size and surface morphological characteristics of the cuprous oxide particles changed noticeably when reduced with separate reducing agents.

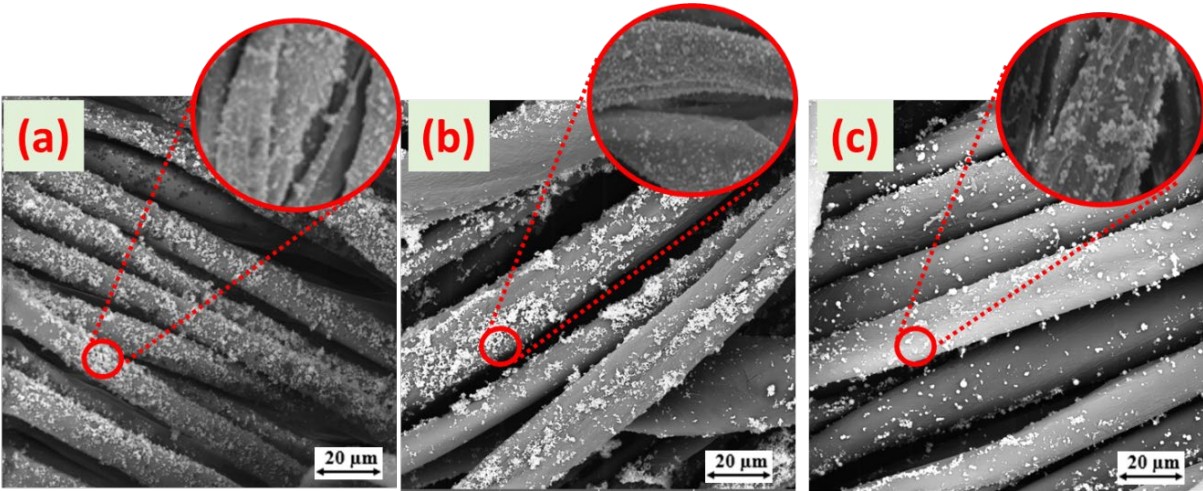

**Figure 7.** Surface morphologies of cotton fabrics coated with cuprous oxide particles reduced with (**a**) DS, (**b**) DA, and (**c**) DG at 100× magnification and their close view at a magnification of 5k×.

It was discovered that the dyed $Cu_2O$ particles-treated fabrics reduced with glucose seemed to have a larger particle size than the dyed $Cu_2O$ particles-treated fabrics (reduced with sodium hydrosulfite and ascorbic acid). In the case of the sodium hydrosulfite, the particles were comparatively small and evenly distributed. The reason is that sodium hydrosulfite is a more powerful and suitable reducing agent for copper salts than glucose and ascorbic acid [14,16]. As shown in Figure 7c, the weak reducing agent (glucose) caused an improper reduction in copper salt, resulting in agglomerated structures that cover much less of the surface of the fiber. The cuprous oxide particles were reduced with ascorbic acid and sodium hydrosulfite and covered the entire surface of the fiber (Figure 7a,b). Figure 7a depicts the uniform and continuous dispersion of particles throughout the surface of the fabric. Furthermore, as the amount of copper salts increased, the deposition became more uniform and denser.

*3.6. XRD Analysis*

The XRD analysis was conducted to determine the phase composition of the deposition of cuprous oxide particles. Figure 8 illustrates the XRD spectrum of the fabric sample in a 2θ range of 10–80 degrees with a 0.02-degree shift. The precise identification of all the diffraction signals from the cuprous oxide structure reveals the phase purity of the produced cuprous oxide nanoparticles. The reflections that were represented by the diffraction peaks at 2θ of 29.6°, 36.5°, 42.4°, 52.1°, 61.5°, and 73.7° were (1 1 0), (1 1 1), (2 0 0), (2 1 1), (2 2 0), and (3 1 1), respectively [30,31]. The sharpness of the signals validated the crystalline character of the cuprous oxide nanoparticles; however, the broadening of the signals supported the production of nanosized cuprous oxide particles. As such, no characteristic peaks of impurities were detected, except the peak of the copper oxide phase at 2θ of 38° [32,33].

*3.7. Antibacterial Analysis*

The antibacterial activity of the copper-treated undyed and dyed fabric samples was tested both qualitatively and quantitatively using standard testing protocols.

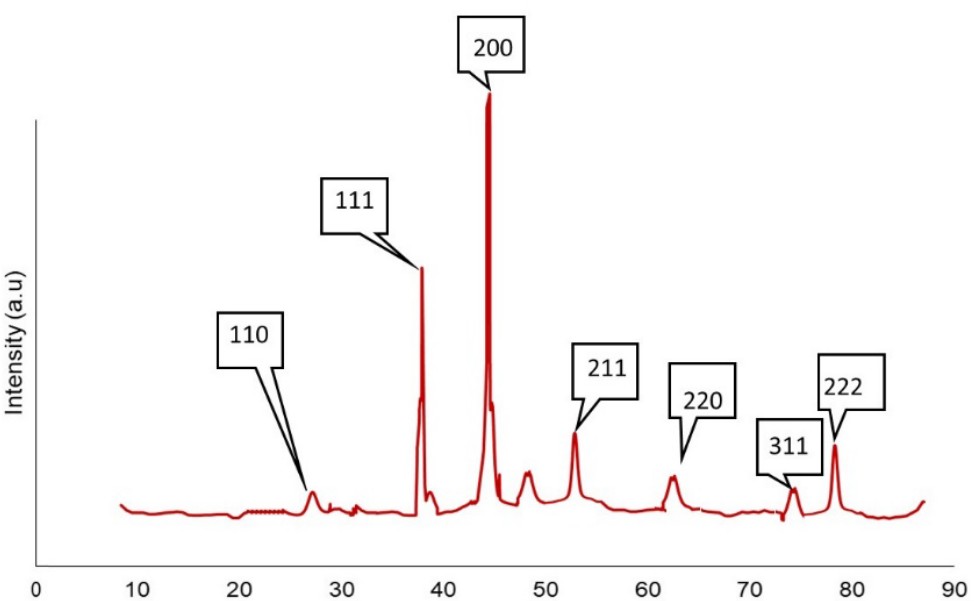

**Figure 8.** XRD patterns of cuprous oxide particles.

### 3.7.1. Zone of Inhibition Tests

The AATCC-147 (disc-diffusion method) standard method was followed for the qualitative assessment of all treated samples. The antibacterial efficacy of all the samples was evaluated against both Gram-negative *(E. coli)* and Gram-positive *(S. aureus)* bacteria. The test was carried out three times, and the mean value calculated for each sample is presented in Table 6. The obvious zones of inhibition around each fabric sample after 24 h of incubation at 37 °C in the dark are depicted in Figures 9 and 10. All the samples (undyed and dyed copper coated fabrics) showed a significant zone of inhibition (ZOI) against both test microbes. However, it was observed that the ZOIs of the dyed copper-coated samples were higher than those of the undyed copper-coated fabrics. The higher values of ZOI of the dyed copper-coated fabrics revealed that the application of the antibacterial dye on the copper-coated fabrics did not mask the antibacterial effect of the copper particles. Rather, the antibacterial effectiveness increased after the dyeing of the copper-treated fabrics.

**Table 6.** The values of zone of inhibitions for developed samples against *S. aureus* and *E. coli*.

| Sr.# | Sample Code | ZOI (mm) | |
| --- | --- | --- | --- |
| | | *S. aureus* | *E. coli* |
| 1. | UT | 0 | 0 |
| 2. | G | 5 | 2 |
| 3. | DG | 7 | 3 |
| 4. | A | 4 | 3 |
| 5. | DA | 7 | 3 |
| 6. | S | 5 | 3 |
| 7. | DS | 8 | 4 |

### 3.7.2. Reduction Factor (Quantitative Test)

The ISO-20743 standard testing protocol was followed for the quantitative evaluation of the antibacterial efficacy of all the developed samples against Gram-positive *(E. coli)* and Gram-negative *(S. aureus)* bacterial strains. The number of inoculated and surviving bacterial colonies was taken, and the percentage reductions were calculated (Table 7). It was observed that all the tested samples exhibited excellent antibacterial potential against both test microbes, i.e., a >85% inhibition of bacterial growth. In case of sample G, A, and S, the maximum antibacterial action was observed in sample S. It was also observed that

the antibacterial activity of all the samples increased significantly after the application of the modified antibacterial dye on the treated fabrics. The activity was increased from 87%, 90%, and 98% to 99% for samples DG, DA, and DS, respectively, against *E. coli*. The same increasing trend was observed for all three samples against *S. aureus*. The increase in antibacterial activity after the application of the dye could be ascribed to the presence of a strong antibacterial agent, i.e., triclosan, which is reported to have excellent antibacterial effects against a broad range of bacterial species. No antibacterial activity was observed for the untreated cotton fabric (UT), which further confirmed that the antibacterial activity in the treated samples was due to the application of the nanoparticles and antibacterial dye.

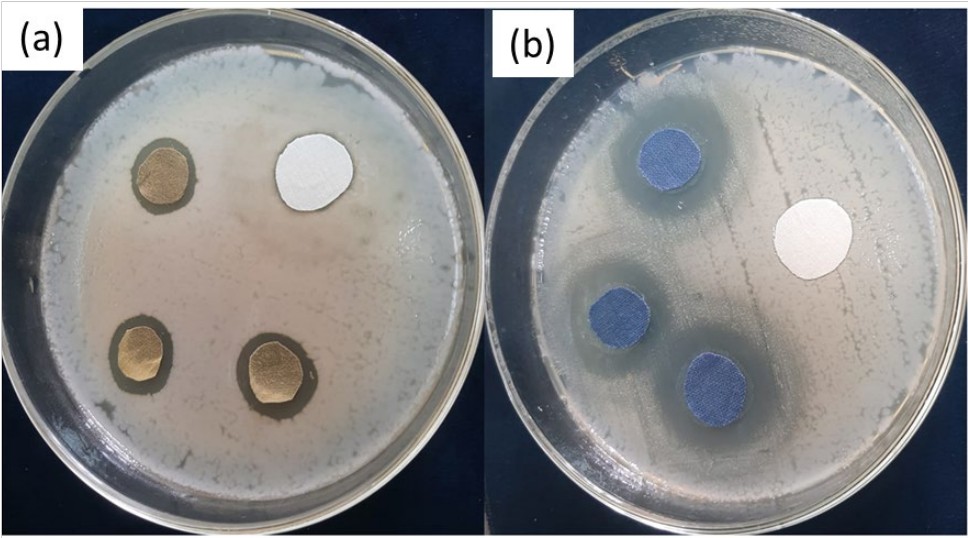

**Figure 9.** ZOI around (**a**) copper-coated fabrics and (**b**) copper-coated dyed fabrics against *E-coli*.

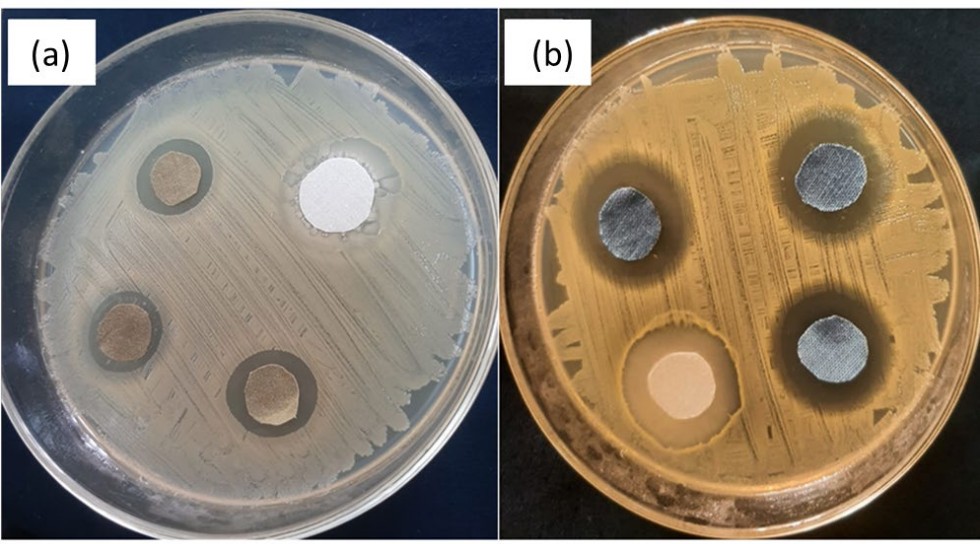

**Figure 10.** ZOI around (**a**) copper-coated fabrics and (**b**) copper-coated dyed fabrics against *S-aureus*.

For more clarification, the reductions in the colony forming units (CFUs/mL) of the surviving bacterial colonies were calculated, and the obtained results are presented in Figure 11. The untreated cotton fabric exhibited a substantial number of surviving bacterial colonies, and higher values of CFUs/mL were obtained (7.34 for *E. coli* and 6.44 for the *S. aureus* bacterial strain). The results revealed that the CFUs concentration was remarkably reduced in all the treated samples. The samples S and DS exhibited the highest reductions

in surviving bacterial colonies, and the CFU values reached 0 from 7.34 and 6.44 for *E. coli* and *S. aureus*, respectively.

**Table 7.** The percentage reductions of developed samples against *S. aureus* and *E. coli*.

| Sr.# | Reducing Agent | Application of Dye | Sample Code | *E. coli* | *S. aureus* |
|---|---|---|---|---|---|
| 1. | Untreated cotton | No | UT | 0% | 0% |
| 2. | Glucose | No | G | 87% | 91% |
| 3. | Glucose | Yes | DG | 99.99% | 97.99% |
| 4. | Ascorbic acid | No | A | 90.99% | 95.99% |
| 5. | Ascorbic acid | Yes | DA | 99.9% | 99.99% |
| 6. | Sodium hydrosulfite | No | S | 98.99% | 99.99% |
| 7. | Sodium hydrosulfite | Yes | DS | 99.99% | 99.99% |

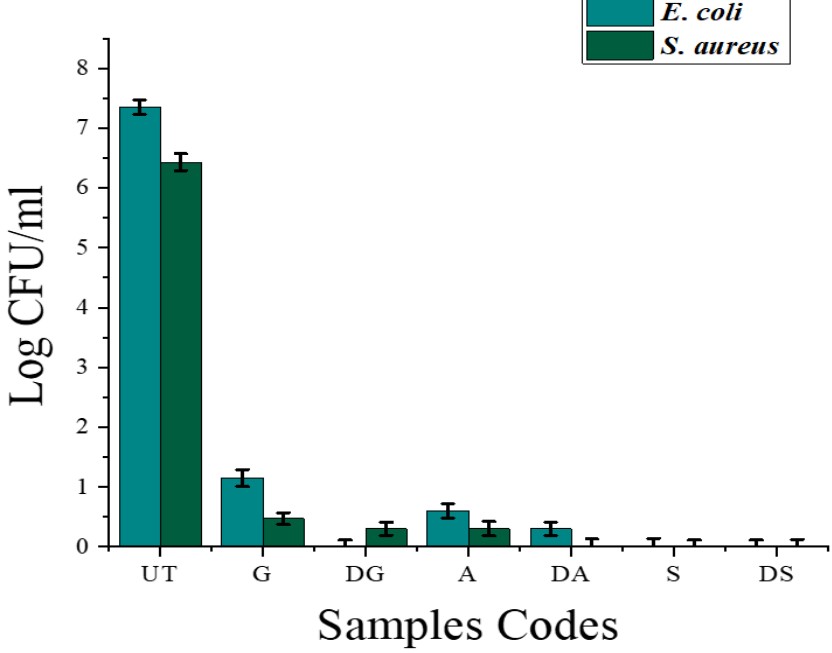

**Figure 11.** The reduction in CFUs of surviving bacterial colonies for all developed samples against *S. aureus* and *E. coli*.

The obtained results were further supported by the selected images provided in Figure 12a,b for untreated and treated samples against *S. aureus* and *E. coli.* An increase in the number of bacterial colonies in the untreated fabric sample was observed, which suggested a complete absence of antibacterial action in the UT sample, whereas, a sharp decrease in the number of bacterial colonies was seen in all the treated samples, suggesting a > 99% reduction in bacterial growth.

### 3.7.3. Mechanism of Antibacterial Action

The combination of both physical and chemical interactions between the bacteria and copper particles is responsible for the antibacterial effect of copper-coated fabrics. The copper NPs can integrate into the cells via endocytic processes. The cellular absorption of ions increased as ionic components were liberated inside the cells via nanoparticle dissociation [34]. As a result, the cell developed a high intracellular concentration that allowed for additional severe oxidative stress leading to microbial cell death. Figure 13

provides a summary of the mechanisms underlying the antibacterial action of copper nanoparticles. The increase in the diameters of the ZOIs of dyed copper-coated samples is due to the presence of triclosan in the dye molecule that has excellent antibacterial action. Triclosan potentially exerts its antibacterial effects by adhering to the surface of bacteria, penetrating the bacterial cell wall, and finally destroying the essential elements critical for bacterial survival, which results in bacterial cell death. Other suggested mechanisms for its antibacterial effect include the inhibition of the coupling between phosphorylation processes and electron transport, the immobilization of essential proteins, and changes in cell membrane permeability. This ultimately prevents ATP generation.

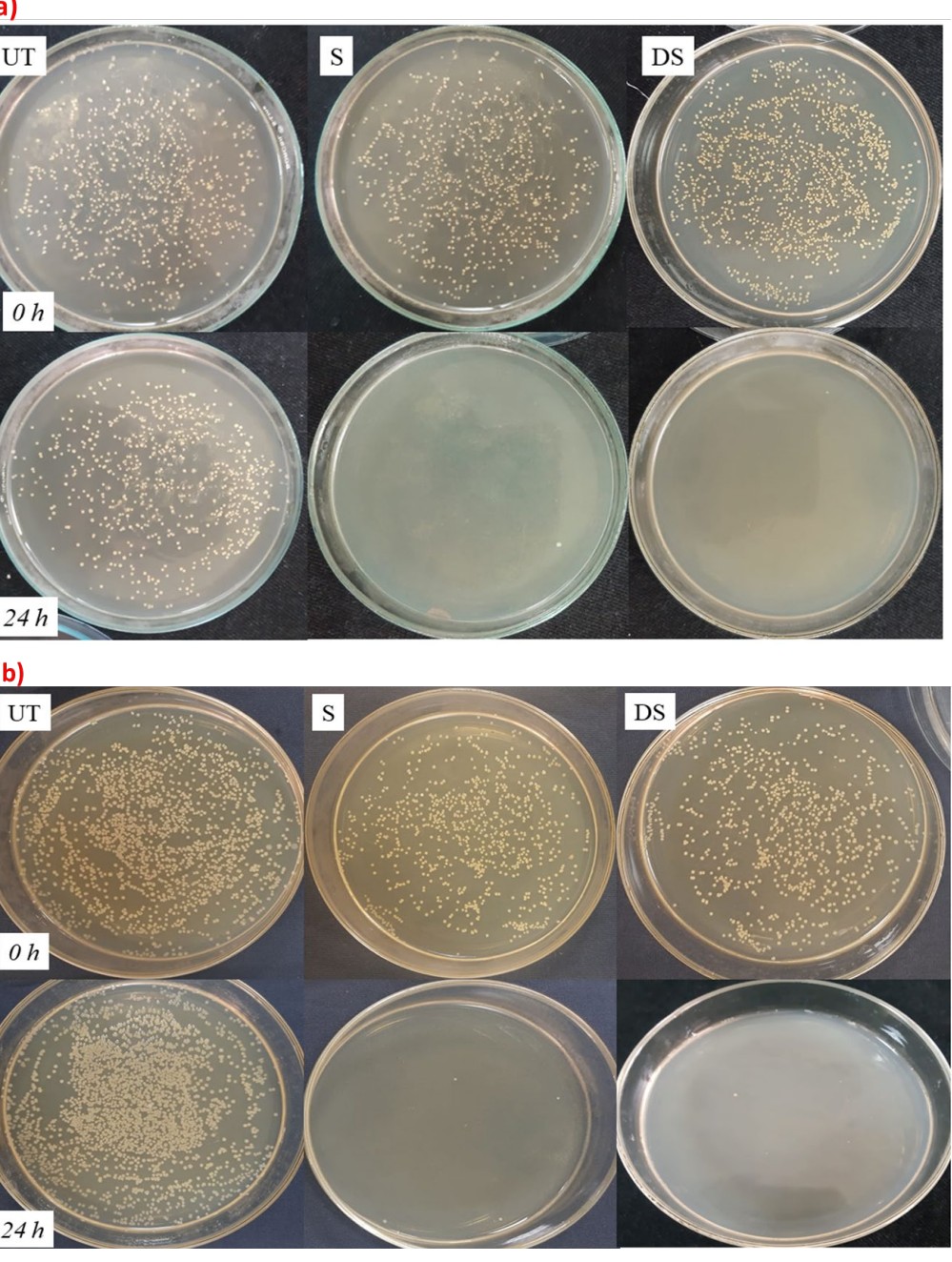

**Figure 12.** (**a**): images for the number of bacterial colonies that were inoculated (0 h) and survived (24 h) for samples UT, S, and DS against *S. aureus.* (**b**): images for the number of bacterial colonies that were inoculated (0 h) and survived (24 h) for samples UT, S, and DS against *E. coli.*

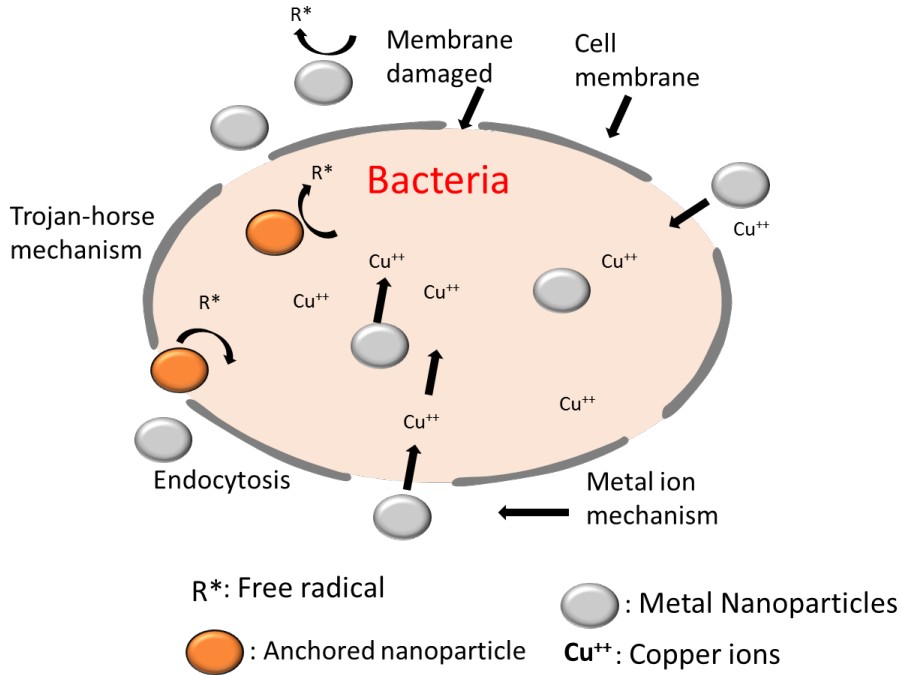

**Figure 13.** A simple overview of the mechanisms involved in the antimicrobial actions of metal nanoparticles, including catalyzed reactive species, metal ion discharge, and the "Trojan horse effect" caused by endocytosis mechanisms [21].

### 3.8. Antifungal Activity of Treated Samples

The AATCC-100 standard method was used for the quantitative evaluation of the antifungal activity against the *A. Niger* fungal specie. The percentage reductions in fungal growth were calculated for all samples, and the obtained results are presented in Table 8. All the treated (treated undyed and treated dyed) fabric samples revealed good antifungal action against the test microbe. The antifungal activity was increased for all the dyed samples, which indicated that the dye ha good antifungal action. However, the antibacterial action of the colored samples was more prominent in comparison with antifungal activity. This might be ascribed to the fact that the antibacterial action of triclosan is higher than its antifungal and antiviral action. Sample S, among all the undyed samples, and sample DS, among all the dyed samples, however, exhibited the highest reductions in fungal spore growth, with maximum antifungal activities of 89% and 91%, respectively. The untreated fabric remained ineffective against the test microbe, which further confirmed that the antifungal action in all the treated samples was due to the application of the nanoparticles and modified antimicrobial dye.

**Table 8.** Reduction percentages in antifungal activity.

| Sr.# | Reducing Agent | Application of Dye | Sample Code | *A. Niger* |
|------|----------------|--------------------|-------------|------------|
| 1. | Untreated cotton | No | UT | 0% |
| 2. | Glucose | No | G | 75% |
| 3. | Glucose | Yes | DG | 79% |
| 4. | Ascorbic acid | No | A | 83% |
| 5. | Ascorbic acid | Yes | DA | 85% |
| 6. | Sodium hydrosulfite | No | S | 89% |
| 7. | Sodium hydrosulfite | Yes | DS | 91% |

### 3.9. Antiviral Effectiveness

Behren and Karber's method was used for the evaluation of virus titer reductions from the initial viral titer of infectivity ($10^8$) against *Corona Virus.* Figure 14 shows the virus infectivity titer log at contact time (0 h and 60 min). Figure 14a,b shows the infectivity titer changes of coronavirus for all the tested samples. It was noticed that antiviral activity increased for all the dyed samples, which indicated that the dye has significant antiviral activity. However, the antibacterial action of the dyed samples was more pronounced compared with their antiviral activity. This could be explained by the fact that the antibacterial action of triclosan is stronger than its antiviral and antifungal action. The observed trend supported the obtained results of the antibacterial activities of these samples. The viral infectivity titer decreased significantly for all the treated samples. However, the maximum reductions were exhibited by sample S (among all the undyed samples) and DS (among all the dyed samples), which showed 79% and 83% antiviral action, respectively. The untreated fabric remained ineffective against the virus, which further confirmed that the antiviral action in all the treated samples was due to the application of the nanoparticles and modified antimicrobial dye. The antiviral action shown by the fabrics treated with copper nanoparticles and the dyed fabric samples could be due to the attachment of the nanoparticles and non-polar part (benzene ring) of the triclosan molecules with glycoproteins on the viral surface, behaving as an inhibitory action for viruses. The reduction percentages of all the tested samples are given in Table 9.

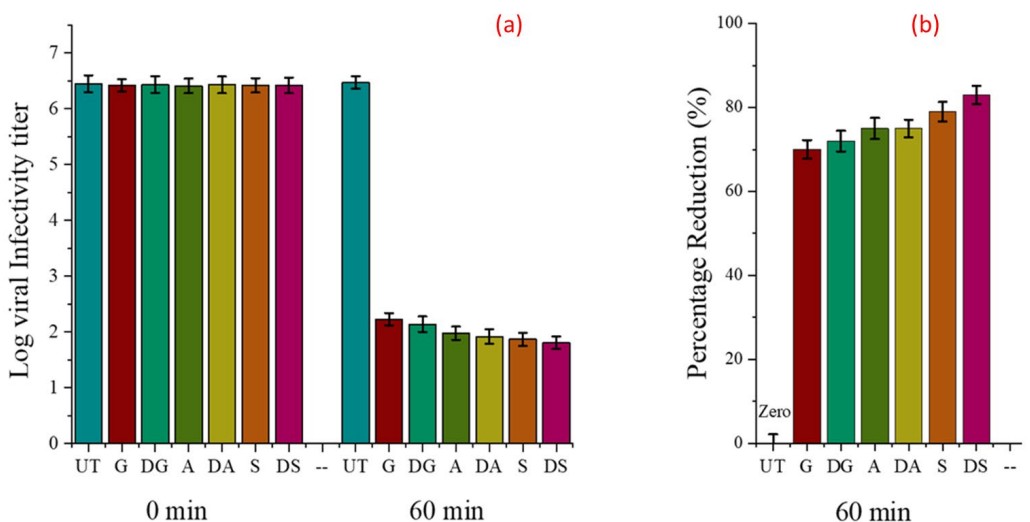

**Figure 14.** Reduction in viral infectivity titer (**a**) calculated from viral infectivity at a contact time of 0 and 60 min, (**b**) and percentage reduction.

**Table 9.** Reduction percentages in antifungal activity.

| Sr.# | Reducing Agent | Application of Dye | Sample Code | Coronavirus |
|------|----------------|--------------------|-------------|-------------|
| 1. | Untreated cotton | No | UT | 0% |
| 2. | Glucose | No | G | 70% |
| 3. | Glucose | Yes | DG | 72% |
| 4. | Ascorbic acid | No | A | 75% |
| 5. | Ascorbic acid | Yes | DA | 75% |
| 6. | Sodium hydrosulfite | No | S | 79% |
| 7. | Sodium hydrosulfite | Yes | DS | 83% |

### 3.10. Durability of Cuprous-oxide-Coated Fabrics

As it was previously mentioned, the copper nanoparticles were attached to the fabric surfaces through different types of physical and chemical linkages. The additional Cu-NPs stacked the microfibers together to construct sterile antibacterial networks by filling up the crevices and voids between them. The durability of the antibacterial activity against repeated washing further indicated this behavior of absorption and adhesion. Additionally, the functionalized fabric samples were squeezed, twisted, and soaked in the water. The samples coated with Cu-NPs showed satisfactory washing characteristics without peeling off from the fabric surface and precipitating into the water. Later, transparent tape was used to conduct an adhesion test. The tape remained transparent because there were no detectable particles on the tape. It confirmed the strong interactions and excellent mechanical adherence characteristics between the Cu-NPs, dye molecules, and fabric.

The antibacterial activities of all the treated fabric samples were examined after washing in order to investigate their durability. All fabric samples were washed following the ISO 105-C01 standard washing test procedure. Table 10 lists the antibacterial values of all the tested samples both before and after washing. The samples were subjected to 20 and 40 laundry cycles. The obtained values make it clear that the values of the inhibition zone changed insignificantly after 20 washing cycles. However, after 40 laundry cycles, there were slight losses in ZOI diameter, which showed that the antibacterial activities were slightly reduced after 40 laundry cycles.

**Table 10.** Antimicrobial properties of $Cu_2O$-coated fabrics after washing.

| Sr.# | Sample Code | ZOI (mm) Unwashed Samples | | ZOI (mm) Washed Samples | | | |
|---|---|---|---|---|---|---|---|
| | | *S. aureus* | *E. coli* | *S. aureus* | | *E. coli* | |
| | | | | 20 Washes | 40 Washes | 20 Washes | 40 Washes |
| 1. | G | 5 | 2 | 5 | 4 | 2 | 2 |
| 2. | DG | 7 | 3 | 6 | 5 | 3 | 2 |
| 3. | A | 4 | 3 | 4 | 3 | 2 | 1 |
| 4. | DA | 7 | 3 | 7 | 5 | 3 | 2 |
| 5. | S | 5 | 3 | 5 | 4 | 2 | 2 |
| 6. | DS | 8 | 4 | 7 | 6 | 4 | 3 |

### 4. Conclusions

Cuprous nanoparticles have been intensively investigated because of their technological aspects and uses in the medical fields. However, some undesirable effects such as discoloration or staining on the particle-coated fabrics were overcome via subsequent dyeing with a bioactive functionalized dye. Six hygienic functionalized textile specimens (three dyed and three undyed) were produced by combining the nanoparticles of cuprous oxide particles with different compositions of three different reducing agents. SEM, dynamic light scattering, FTIR (FTIR Perkin-Elmer, Liberec, Czech Republic), EDS (Quanta 50, FEI, Liberec, Czech Republic), and XRD (Malvern Panalytical's X-ray diffraction, Liberec, Czech Republic) were used to examine the surface morphologies and metal presences. After that, the Reactive Blue 4 dye was functionalized with triclosan to impart antibacterial activity to the dye. The FTIR results confirmed the successful modification of the dye with the antibacterial agent. The modified dye was applied on copper-treated cotton fabrics with the exhaust dyeing method. The modified dye exhibited excellent fixation, exhaustion, and dye levelness on the copper-treated fabrics. The antibacterial activity of the copper-treated fabrics was also increased after the application of the dye.

The cuprous-oxide-treated dyed and control fabrics were tested for antibacterial activity using quantitative and qualitative measurements. It was observed that the ZOIs of the dyed copper-coated samples were higher than those of the undyed copper-coated fabrics. The strongest antibacterial effect was found for the dyed fabric sample DS (sodium hydrosulfite). In the case of the quantitative analysis, samples S and DS exhibited the

highest reductions in surviving bacterial colonies, and the CFUs values reached 0 from 7.34 and 6.44 for *E. coli* and *S. aureus*, respectively.

However, the antibacterial action of the colored samples was more prominent in comparison with their antifungal activity. This might be ascribed to the fact that the antibacterial action of triclosan is higher than its antifungal and antiviral action. Furthermore, the antimicrobial effects and SEM images after washing were used to affirm the durability of the deposition. The reliability of the particles on the fabrics' surfaces (as seen in the SEM images) confirmed that the particles were firmly attached to the fibers and interspaces. Moreover, the changes in antimicrobial activities for all the treated fabric samples after repeated laundry cycles were insignificant, which further confirmed the durability of the particles on the fabrics. The developed process is simple, low-cost, and provides odorless work wear. The effective utilization of cuprous-oxide-coated fabrics demonstrated their potential applications in the area of medical textiles, such as in the developments of antimicrobial surgical curtains, trousers, boots, panels, bedspreads, surgical gowns, drapes, panel covers, chair and table covers, patient and doctors' socks, and so on.

**Author Contributions:** Conceptualization, J.W.; Methodology, A.A. and S.P.; software, A.A.; Validation, S.P.; formal analysis, J.M.; resources, S.P.;Investigation, N.Z. and M.S.A.; Data curation, M.S. and S.P.; writing—review and editing, A.A.; visualization, B.T.; supervision, B.T.; Project administration, B.T.; Funding acquisition, J.M. and S.P. All authors have read and agreed to the published version of the manuscript.

**Funding:** This research was supported by the project "Textile structures combining virus protection and comfort" (reg.c.:cz.01.1.02/0.0/0.0//20_321/0024467). This work was also supported by the project "Advanced structures for thermal insulation in extreme conditions" (Reg. No. 21-32510 M) granted by the Czech Science Foundation (GACR).

**Institutional Review Board Statement:** Not applicable.

**Informed Consent Statement:** Not applicable.

**Data Availability Statement:** Not applicable.

**Conflicts of Interest:** The authors declare no conflict of interest.

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
