# Peer review of "Copper-Treated Environmentally Friendly Antipathogenic Cotton Fabric with Modified Reactive Blue 4 Dye to Improve Its Antibacterial and Aesthetic Properties"

_coatings, doi:10.3390/coatings13010133_

Round 1
Reviewer 1 Report
Antimicrobial cotton fabric is promising for the applications in bed sheets, surgical drapes, panels, surgical gowns, pants, panel covers, wall papers coverings, etc. The topic of this manuscript is of broad interest to the readers. However, major revision is required before acceptation.
1. “a reactive dye was made antibacterial functional dye by reacting…” in line 17 needs to be revised.
2. What cause the different antipathogenic activity of cuprous oxide particles synthesized from three different types of reducing agents? Particle size or crystals?
3. “the current study has proposed a novel approach for the development of copper-coated antibacterial cotton fabric…” in line 75-76 should be revised since Cu2O other than copper is coated. Please double check the whole manuscript to revise such expressions.
4. A full stop should be added for each point listed in line 85 to 93.
5. How about the stability of Cu2O layer? Does it will transfer into CuO?
6. Please double check the whole manuscript to remove typos of subscripts and superscripts, for example “220 g/m2” in line 102 and “Na2CO3” in Figure 1 need to be revised.
7. The authors should pay attention to the writing of units. There should be a space between the unit and the number.
8. SEM images at higher magnification are suggested to be added to show the particle size difference.
9. XRD patterns of cuprous oxide particles prepared from three different types of reducing agents are suggested to be added to compare the difference.
10. Most of the references are too old which cannot represent the update progress in the field. More references published in recently years are suggested to be added, for example Journal of Bioresources and Bioproducts 2021, 6 (1), 75-81; Journal of Bioresources and Bioproducts 2021, 6 (1), 26-32; Journal of Bioresources and Bioproducts 2020, 5 (3), 180-185.
Author Response
Reviewer 1
- “a reactive dye was made antibacterial functional dye by reacting…” in line 17 needs to be revised.
Answer: revised in text and highlighted as red. Thanks
- What cause the different antipathogenic activity of cuprous oxide particles synthesized from three different types of reducing agents? Particle size or crystals?
Answer: the antibactrerial activity depends upon the maximum number of cuprous oxide ions. The maximum reduction of copper salt was achieved in case of sodium hydrosulphite. Secondly. there was obvious change in size and surface morphology of cuprous oxide particles reduced by different reducing agents. It was noticed that the copper salt reduced by glucose, having the big particle size as compared to cuprous oxide particles coated fabrics (reduced by ascorbic acid and Sodium hydrosulphite). While the comparatively smallest and even distribution of particles was observed in case of sodium hydrosulphite. The reason is that the sodium hydrosulphite is strongest and more compatible reducing agent for copper salts as compared to the ascorbic acid and glucose [14] [16]. The weak reducing agent (glucose) provided the improper reduction of copper salt and produced the agglomerated structures, which in turn covers the less surface of fibre as shown in Figure 8c. The cuprous oxide particles reduced by sodium hydrosulphite and ascorbic acid covered the complete fibre surface (Figure 8b and 8a). The figure 8a showed the continuous and uniform distribution of particles on the surface of cotton. Furthermore, the deposition was found more uniform and denser with increase in concentration of copper salts. Hence, more coverage of surface of fabric by particles (in case of sodium hydrosulphite) will show more effect against pathogens.
Added in text
- “the current study has proposed a novel approach for the development of copper-coated antibacterial cotton fabric…” in line 75-76 should be revised since Cu2O other than copper is coated. Please double check the whole manuscript to revise such expressions.
Answer: revised in text and highlighted as red. Thanks
- A full stop should be added for each point listed in line 85 to 93.
Answer: Revised in text and highlighted as red. Thanks
- How about the stability of Cu2O layer? Does it will transfer into CuO?
Response: The Cu was properly reduced to Cu2O on the fabric. Moreover, aging properties of developed samples were studied after repeated launderings to further confirm the stability of Cu2O. The aging properties were remained unchanged even after severe launderings which established that Cu2O layer is highly stable and is not oxidized to CuO. (text is also added in the manuscript)
- Please double check the whole manuscript to remove typos of subscripts and superscripts, for example “220 g/m2” in line 102 and “Na2CO3” in Figure 1 need to be revised.
Response: Corrected. thanks
- The authors should pay attention to the writing of units. There should be a space between the unit and the number.
Response: Corrected.
- SEM images at higher magnification are suggested to be added to show the particle size difference.
Response: Answer: Close views are also added in SEM analysis. Thanks
- XRD patterns of cuprous oxide particles prepared from three different types of reducing agents are suggested to be added to compare the difference.
Response: Thank you for your suggestion bur XRD is not currently available. XRD pattern of nanoparticles prepared from different reducing agents will be different with respect to the peak sharpness. The prime objective of the XRD was to confirm that Cu2O is formed. Moreover, the antibacterial activity of all the particles were same. Therefore, this discussion would be irrelevant and paper is already too lengthy.
- Most of the references are too old which cannot represent the update progress in the field. More references published in recently years are suggested to be added, for example Journal of Bioresources and Bioproducts 2021, 6 (1), 75-81; Journal of Bioresources and Bioproducts 2021, 6 (1), 26-32; Journal of Bioresources and Bioproducts 2020, 5 (3), 180-185.
Response: References are updated.
Reviewer 2 Report
The paper is interesting and falls within the scope of the journal. The authors should explore more systematically the effect of repetitive washing cycles on the durability of the coatings. What are the limitations of this approach?
Author Response
Reviewer 2
The paper is interesting and falls within the scope of the journal. The authors should explore more systematically the effect of repetitive washing cycles on the durability of the coatings. What are the limitations of this approach?
Response: The effect of repetitive washing cycles on the durability of the coatings is explained systematically as per your suggestion.
Reviewer 3 Report
The manuscript by Azam and co-workers reports three methods for depositing copper oxide nanoparticles onto cotton fibers. The resulting materials were tested for their antibacterial properties. A modified dye was deposited on the same materials. The influence of the dye on the antibacterial properties of these materials and their color was investigated. The resulting materials have been shown to have significant antibacterial properties that persist even after they have been washed.
The topic of the study is relevant in the field and its motivation is justified. The conclusions drawn are consistent with the results obtained. Cited references relate to the research topic and are justified. The objectives, experimental methods, and main findings of the study are clearly presented.
However, the manuscript has a number of critical flaws.
1. The 13C NMR spectrum in Figure 7 is unrelated to the triclosan functionalized dye Reactive Blue 4. The signal assignment is wrong.
2. It cannot be deduced from the study that the antibacterial properties of cotton fabric cannot be guaranteed by the modified dye alone and that the presence of nanoparticles is absolutely necessary.
3. The meaning of some phrases is not clear:
“At first, fabrics were sensitized with citric acid then the formation of Cu2O particles was done by Fehling solution method. Then, the cuprous oxide particles were deposited on cotton fabrics. In second step, a reactive dye was made antibacterial functional dye by reacting the Reactive Blue 4 dye with triclosan (antibacterial agent).”
“It was noticed that the dyed cuprous oxide particles coated fabrics, which were reduced by glucose, having the big particle size as compared to the dyed cuprous oxide particles coated fabrics (reduced by ascorbic acid and Sodium hydrosulphite). While the comparatively smallest and even distribution of particles was observed in case of sodium hydrosulphite.”
Author Response
Reviewer 3
- The 13C NMR spectrum in Figure 7 is unrelated to the triclosan functionalized dye Reactive Blue 4. The signal assignment is wrong.
Response: Signal assignment is corrected and supported with reference in response your kind suggestion.
- It cannot be deduced from the study that the antibacterial properties of cotton fabric cannot be guaranteed by the modified dye alone and that the presence of nanoparticles is absolutely necessary.
Response: The fabric treated with antibacterial dyes have poor washing durability due to which their application on fabric is not sustainable. Therefore, the synthesis and durable immobilization of nanoparticles particularly copper nanoparticles on textiles gained considerable attention in recent years due to their excellent washing durability. However, the textiles treated copper or silver nanoparticles significantly alter the hue of the fabric thus affecting its aesthetic properties. Therefore, in the current study, the nanoparticles were applied on cotton fabric to achieve durable antimicrobial activity and the treated fabric were dyed with the antibacterial dye to maintain its aesthetic as well as antibacterial properties. (text is also added in manuscript)
- The meaning of some phrases is not clear
“At first, fabrics were sensitized with citric acid then the formation of Cu2O particles was done by Fehling solution method. Then, the cuprous oxide particles were deposited on cotton fabrics. In second step, a reactive dye was made antibacterial functional dye by reacting the Reactive Blue 4 dye with triclosan (antibacterial agent).”
“It was noticed that the dyed cuprous oxide particles coated fabrics, which were reduced by glucose, having the big particle size as compared to the dyed cuprous oxide particles coated fabrics (reduced by ascorbic acid and Sodium hydrosulphite). While the comparatively smallest and even distribution of particles was observed in case of sodium hydrosulphite.
Response: The mentioned phrases are corrected.
Reviewer 4 Report
First, thank you for letting me the opportunity to send some comments to improve your manuscript.
1. The manuscript say "the objectives of the present study were to develop the environmental friendly, low price, easy and fast method for developing the antipathogenic (antibacterial, antifungal and antiviral) cuprous oxide coated multifunctional fabrics" which is not too related to the title of the manuscript. Please review and write the article title more close to the objective.
2. The phrase "During the COVID-19 pandemic, the prominence of antimicrobial textiles primarily those used in hospitals has escalated" need a cite to support it.
3. For the phrase "Recently, metallic nanoparticles are increasingly being used to functionalize fabrics 43 in an effort to prevent the transmission of disease and bacterial growth" please include a citation.
4. In the phrase "The objectives of the present research are:" are including some processes and not only objectives. Please review and write only research objective.
5. Please include the next references:
https://doi.org/10.1016/j.fct.2022.113366
https://doi.org/10.3390/pr10061207
https://doi.org/10.1016/j.colsurfa.2022.128978
https://doi.org/10.3390/pr10071308
https://doi.org/10.3390/pr10091727
https://doi.org/10.1016/j.carbpol.2019.01.066
https://doi.org/10.3390/pr10061088
https://doi.org/10.3390/molecules25245802
https://doi.org/10.3390/polym13121906
6. Line 102 must say "f 220 g/m2."
7. It can be more clear to describe the 2.2 Preparation using a fluxogram.
8. In Figure 1 the Na2CO3 must be write as Na2CO3
9. The reference must include DOIs of all the articles.
Author Response
Reviewer 4
1. The manuscript say "the objectives of the present study were to develop the environmental friendly, low price, easy and fast method for developing the antipathogenic (antibacterial, antifungal and antiviral) cuprous oxide coated multifunctional fabrics" which is not too related to the title of the manuscript. Please review and write the article title more close to the objective.
Answer: Title is changed
Copper treated environmental friendly antipathogenic cotton fabric with modified reactive blue 4 dye to improve its anti-bacterial and aesthetic properties
The phrase "During the COVID-19 pandemic, the prominence of antimicrobial textiles primarily those used in hospitals has escalated" need a cite to support it.
Answer: Reference is added.
3. For the phrase "Recently, metallic nanoparticles are increasingly being used to functionalize fabrics 43 in an effort to prevent the transmission of disease and bacterial growth" please include a citation.
Answer: Reference is added.
4. In the phrase "The objectives of the present research are:" are including some processes and not only objectives. Please review and write only research objective.
Answer: We have replaced the un necessary lines from objectives. Thanks
5. Please include the next references:
https://doi.org/10.1016/j.fct.2022.113366
https://doi.org/10.3390/pr10061207
https://doi.org/10.1016/j.colsurfa.2022.128978
https://doi.org/10.3390/pr10071308
https://doi.org/10.3390/pr10091727
https://doi.org/10.1016/j.carbpol.2019.01.066
https://doi.org/10.3390/pr10061088
https://doi.org/10.3390/molecules25245802
https://doi.org/10.3390/polym13121906
Answer: 6 references are cited.
6. Line 102 must say "f 220 g/m2."
Answer: Done
7. It can be more clear to describe the 2.2 Preparation using a fluxogram.
Answer: The flow chart for the preparation of cuprous oxide particles particles and their deposition on cotton fabric is already given in Figure 3. Thanks
In Figure 1 the Na2CO3 must be write as Na2CO3
Answer: Corrected.
Round 2
Reviewer 1 Report
The manuscript has been well revised according to the comments and could be accepted.
Author Response
Thank you for all your worthy and kind suggestions and for approving all the revisions.
Reviewer 3 Report
The 13C NMR spectrum in Figure 7 cannot belong to the triclosan functionalized dye Reactive Blue 4. The signal assignment is incorrect. There are no saturated carbons in anthracene. The number of chemically different aromatic carbons must be larger. Carbonyl carbons resonate at 180 ppm.
Author Response
Response: Spectra was removed and FTIR results are enough. thanks